# Interventions to increase early infant diagnosis of HIV infection: A systematic review and meta-analysis

**Babasola Okusanya**[1]*, **Linda J. Kimaru**[1], **Namoonga Mantina**[1], **Lynn B. Gerald**[1‡], **Sydney Pettygrove**[2‡], **Douglas Taren**[1‡], **John Ehiri**[1]

**1** Department of Health Promotion Sciences, Mel and Enid Zuckerman College of Public Health, University of Arizona, Tucson, Arizona, United States of America, **2** Department of Epidemiology, Mel and Enid Zuckerman College of Public Health, University of Arizona, Tucson, Arizona, United States of America

☯ These authors contributed equally to this work.
‡ LBG, SP and DT also contributed equally to this work.
* bokusanya@email.arizona.edu

## Abstract

### Objectives

Early infant diagnosis (EID) of HIV infection increases antiretroviral therapy initiation, which reduces pediatric HIV-related morbidity and mortality. This review aims to critically appraise the effects of interventions to increase uptake of early infant diagnosis.

### Design

This is a systematic review and meta-analysis of interventions to increase the EID of HIV infection. We searched PubMed, EMBASE, CINAHL, and PsycINFO to identify eligible studies from inception of these databases to June 18, 2020. EID Uptake at 4–8 weeks of age was primary outcome assessed by the review. We conducted meta-analysis, using data from reports of included studies. The measure of the effect of dichotomous data was odds ratios (OR), with a 95% confidence interval. The grading of recommendations assessment, development, and evaluation (GRADE) approach was used to assess quality of evidence.

### Settings

The review was not limited by time of publication or setting in which the studies conducted.

### Participants

HIV-exposed infants were participants.

### Results

Database search and review of reference lists yielded 923 unique titles, out of which 16 studies involving 13,822 HIV exposed infants (HEI) were eligible for inclusion in the review. Included studies were published between 2014 and 2019 from Kenya, Nigeria, Uganda, South Africa, Zambia, and India. Of the 16 included studies, nine (experimental) and seven

**Data Availability Statement:** All relevant data are within the paper and its Supporting Information files.

**Funding:** The author(s) received no specific funding for this work.

**Competing interests:** The authors have declared that no competing interests exist.

**Abbreviations:** ANC, Antenatal care; ART, Antiretroviral therapy; CBT, Cognitive Behavioral Therapy; EID, Early infant diagnosis; EPOC, Effective Practice of Care; GRADE, Grading of recommendations assessment, development, and evaluation; HEI, HIV-exposed infants; HIV, Human Immunodeficiency Virus; PMTCT, Prevention of mother-to-child transmission; RCT, Randomised Controlled Trial; SMS, Short message Systems; SSA, Sub-Saharan Africa.

(observational) studies included had low to moderate risk of bias. The studies evaluated eHealth services (n = 6), service improvement (n = 4), service integration (n = 2), behavioral interventions (n = 3), and male partner involvement (n = 1). Overall, there was no evidence that any of the evaluated interventions, including eHealth, health systems improvements, integration of EID, conditional cash transfer, mother-to-mother support, or partner (male) involvement, was effective in increasing uptake of EID at 4–8 weeks of age. There was also no evidence that any intervention was effective in increasing HIV-infected infants' identification at 4–8 weeks of age.

## Conclusions

There is limited evidence to support the hypothesis that interventions implemented to increase uptake of EID were effective at 4–8 weeks of life. Further research is required to identify effective interventions that increase early infant diagnosis of HIV at 4–8 weeks of age.

## Prospero number

(CRD42020191738).

## Introduction

Globally, 180,000 new pediatric HIV infections occurred in 2017 [1]. Most of these infections occurred in low- and middle-income countries, with India being the only country outside sub-Saharan Africa (SSA) with a high prevalence of HIV infection. In SSA, South Africa and Nigeria are the leading countries contributing to new pediatric HIV infection [2]. Strategies for preventing mother-to-child transmission (PMTCT) of HIV have been implemented to reduce the burden of new pediatric HIV infection. PMTCT strategies aim to achieve an overall pediatric HIV infection rate of less than 5%, from 45% in mothers who breastfeed and a much lower transmission rate of 2% without breastfeeding [3, 4].

The effectiveness of PMTCT is measured by EID of HIV using Deoxynucleic acid (DNA) polymerase chain reaction (PCR) techniques at 4–6 weeks of age [5]. However, there is a low uptake of EID in many settings with high HIV infection prevalence [6, 7]. This uptake may be due to individual maternal factors, health system factors, and community-level factors. EID coverage in Angola, Burundi, Chad, the Democratic Republic of the Congo, and Nigeria is below 15%, and in Malawi, less than 20% of HIV-exposed infants had EID within eight weeks of birth [6, 8]. Despite the Global Plan to achieve a 60% reduction in new pediatric HIV infection between 2009 and 2015 [9], an estimated 1.4 million new pediatric HIV infections occurred in 2017 [10]. Two systematic reviews on PMTCT have evaluated interventions that might increase EID uptake. One of the reviews was of low methodical quality because it included data from conference abstracts and combined observational studies and randomized control trials in meta-analyses [11]. It also included EID performed up to 18 months [11]. The other review included a "Before and After" study with no pre-intervention data for EID uptake [12].

For mothers' convenience, EID is scheduled for the same visit as the maternal postnatal clinic visit six weeks after birth. However, studies have found that women either did not return with their children (46.3%) or received their children's EID results late (46.6%) [7, 13].

Strategies implemented to reduce challenges associated with performing EID and improving HIV-exposed infants' health include integrating EID into immunization clinics, health system improvement strategies, eHealth technologies, and behavioral intervention programs. The use of mobile phone text messaging has been shown to reduce EID reports' turnaround time to health facilities and caregivers to facilitate ART initiation in Zambia [14, 15]. Evidence from an earlier systematic review suggested that EID may have beneficial effect when integrated into routine immunization clinics in rural health facilities [16]. While direct accompaniment of postpartum HIV-positive women to EID testing led to an earlier infant testing in Mozambique, cognitive behavior therapy (CBT) intervention for pregnant HIV positive women in South Africa led to lower EID uptake [17, 18].

Untreated infants living with HIV manifest illnesses early, with about half dying by age 2 and 80% dying before age five [19]. EID at six weeks increased the number of infants diagnosed as living with HIV by threefold [20]. In a simulated model, compared to no testing, EID at six weeks increased the life expectancy of children living with HIV to 61.4 years. It conferred a $1250/year of life saved of the incremental cost-effectiveness ratio [21]. EID reduces the likelihood of viral rebound and increases the probability of survival 7-fold [13, 22]. Thus, this review's objective is to critically appraise and summarize the effect of interventions to increase uptake of early infant diagnosis at 4–8 weeks of age.

## Methods

### Search strategy and selection criteria

An electronic database search of PubMed, EMBASE, CINAHL, and PsycINFO was conducted from inception of the databases to June 18, 2020, with no geographic or language restrictions. The search strategies presented in the S1 Table in yielded publications from 1990 to 2020. The review focused on interventions to increase uptake of EID of HIV among HIV-exposed infants. The review was not limited by time of publication or setting in which the studies conducted. Eligible interventions included eHealth systems Interventions (e.g., short message systems (SMS); Mobile phone calls; other electronic reminder/recall systems), health system improvement (e.g., Tracking of mother/HIV-exposed infants; referral systems; Quality improvement programs; sample transportation facilitation, Direct accompaniment of mother-infant pair to EID testing facility; patient-centered care), integration of EID into maternal and child health other services (e.g., child immunization or infant welfare clinics), behavioral interventions (e.g., conditional cash transfer; Cognitive behavior therapy, Mother-to-Mother (Mentor-Mother)), and any other interventions. The comparison was usual care defined as the return of the HIV exposed infant (HEI) for EID at the six-week postpartum visit or as implemented by the country. Birth testing of HIV exposed infants was an exclusion criterion.

The review's primary outcomes were uptake of early infant diagnosis at 4–8 weeks of age and the identification of HIV-infected infants. Secondary outcomes included 1). Turnaround time of EID result to the caregiver, 2). Turnaround time of EID result to the mother, and 3). Initiation of antiretroviral therapy for an HIV-positive infant. Other information sought include study identity, publication country, study design, participants' characteristics, and a description of the intervention and comparison.

Database search outputs and data extraction were managed with Covidence Systematic Review Software [23]. Three review authors screened eligible studies for inclusion. While half of the potentially eligible studies was screened by BO and LK, the other half was screened by BO and NM. Further, for each pair of reviewers, a third reviewer resolved discordant screening. Eligible studies included randomized controlled studies (individual and cluster RCT),

controlled "before-and-after studies, case-control studies, and cohort studies. The study selection followed the PRISMA guideline as shown in Fig 1 [24].

## Data analysis

Two review authors (BO and LK) independently extracted data on HIV-exposed infants from included studies using Covidence. We resolved differences through discussion and consensus among the reviewers. Where necessary, we contacted authors of eligible studies for additional study information on disaggregated data on review outcomes. We excluded studies when the authors did not provide the requested information within one month, and it was impossible to extract data on the occurrence of EID within 4–8 weeks of age in studies that reported EID uptake beyond 8 weeks of life. Three review authors performed risk of bias assessment of included studies, with half of the risk of bias assessment performed by BO and LK, the other half by BO and NM. The assessments were conducted, using the Cochrane Collaboration's Effective Practice and Organizational Care (EPOC) risk of bias assessment tool [25]. The EPOC tool has nine domains and best fits a systematic review of health promotion and public health interventions [26].

Most perinatal HIV infection transmission and new pediatric HIV infection occur in low- and middle-income countries, particularly SSA [1]. We included non-randomized studies in this systematic review to account for all interventions in SSA. Non-randomized studies were included in meta-analyses only if they were of the same study design and reported on the same outcomes. Pre-intervention and post-intervention data from quasi-experimental studies were

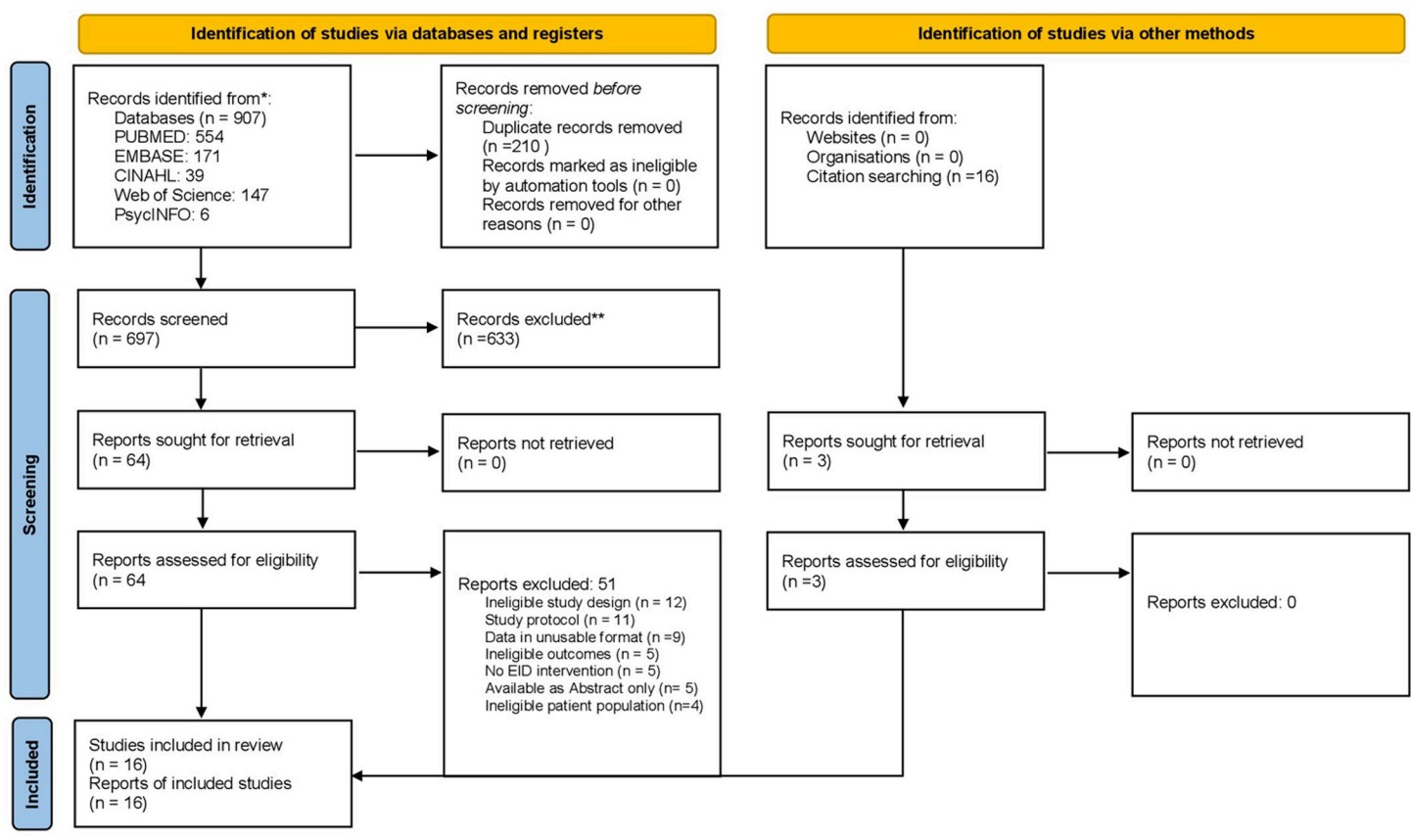

**Fig 1. PRISMA diagram of study selection.**

regarded as comparison and intervention data, respectively. They were included in meta-analyses if they implemented the same interventions. Otherwise, the individual study's effect estimates and confidence intervals were reported. Data from randomized studies were included in a meta-analysis if they reported the same outcomes. Since cluster randomized trials reported individual participant data, they were included in the meta-analysis with individually randomized studies.

Odds ratios (OR) with 95% confidence interval was the measure of effect for dichotomous data. Because of the skewness of continuous outcome data, it was presented as median and interquartile range. As recommended by the Cochrane Collaboration, a meta-analysis of this outcome was not performed [27]. Meta-analysis was conducted per-protocol analysis because studies excluded pregnancies that ended in miscarriage or poor perinatal outcomes. The $I^2$ statistic was used to determine statistical heterogeneity, with an $I^2$ of 50% regarded as substantial heterogeneity [27]. The summary of findings was presented as Forest plots, using a random-effect model for all analyses because of statistical heterogeneity in the meta-analyses.

Further investigations of statistical heterogeneity were conducted with sensitivity analyses for primary outcomes, based on study design (cluster RCT vs Individual RCT) and type of intervention (eHealth interventions vs SMS only intervention) as recommended by the Cochrane handbook for systematic reviews of interventions [27]. Published reports were used for meta-analysis conducted with the Review Manager (Revman; version 5.4.1). Although we had proposed sub-group analyses for primary outcomes based on laboratory-based versus point-of-care (POC) EID testing and EID results turnaround time to caregivers of ≤ 4 weeks versus ≥5 weeks, this could not be done because of few included studies, with none using POC for EID.

The GRADE approach was used to evaluate the quality of evidence [28]. This review was registered on PROSPERO ID: (CRD42020191738). Institutional ethics committee approval was not required because of the use of published reports. The systematic review was completed in December 2020.

### Role of the funding source

There was no funding source for this review.

## Results

The database search yielded 907 titles, and 16 additional studies were identified from the reference lists of eligible studies. After de-publication and screening of titles and abstracts, the full texts of 67 studies were reviewed for inclusion. We excluded 51 studies from the systematic review (See S2 Table). Fig 1. PRISMA flow diagram provides details of the study selection process.

Sixteen studies involving 13,822 HIV-exposed infants (HEI) were included. The studies were published between 2014 and 2019 from Kenya [29–34], Nigeria [35–37], Uganda [38], South Africa [39–41], and Zambia [42]. One study was conducted in India [43]. Of the included studies, nine were of experimental design; five cluster randomized controlled trials (RCTs) [29, 31, 34, 35, 41], three individually randomized controlled trials [33, 36, 44], and one step-wedged cluster RCT [32]. Also, seven observational studies were included; five "Before-and-after studies [30, 38, 40, 42, 43], and two cohort studies [37, 39]. Table 1 summarizes included studies, and the S3 Table in (Characteristics of included studies) provides details of each included study.

**Table 1. Summary of included studies.**

| Author & year | Country of intervention | Study design | Number of participants (Infants ≤ 8weeks of age) | Intervention(s) | Results of pre-specified outcome measures in included studies |
|---|---|---|---|---|---|
| eHealth Interventions | | | | | |
| Coleman 2017 [39] | South Africa | Retrospective cohort study | 639 | MAMA SMS on healthy eating, reminders of ANC/PNC appointments, | EID uptake at 4-8wks: • Intervention: 156/192 • Usual care: 337/447 |
| | | | | psycho-social support, delivery planning, PCR testing reminders, adherence to AR | Positive EID result at 4-8wks: • Intervention: 0/156 • Usual care: 3/337 |
| Kassaye 2016 [31] | Kenya | Cluster RCT | 470 | SMS text messages (3-6/ week) in the local language | EID uptake at 4-8wks: • Intervention: 213/242 • Usual care: 202/228 |
| | | | | | Positive EID result at 4-8wks: • Intervention: 1/213 • Usual care: 3/202 |
| Odeny 2014 [44] | Kenya | RCT | 368 | Individually tailored, theory-based two-way SMS | EID uptake at 4-8wks: • Intervention: 172/187 • Usual care: 154/181 |
| | | | | | Positive EID result at 4-8wks: • Intervention: 2/172 • Usual care: 3/154 |
| Odeny 2019 [32] | Kenya | Cluster RCT | 2326 | Individually-tailored, theory-based two-way SMS | EID uptake at 4-8wks: • Intervention: 1466/1613 • Usual care: 609/713 |
| Sarna 2019 [33] | Kenya | RCT | 309 | A structured, counsellor-delivered, tailored cell phone counseling | EID uptake at 4-8wks: • Intervention: 145/181 • Usual care: 104/128 |
| | | | | | Positive EID result at 4-8wks: • Intervention: 7/127 • Usual care: 2/181 |
| Schwartz 2015 [40] | South Africa | Before and After | 99 | Weekly SMS messages until 6 weeks postpartum | EID uptake at 4-8wks: • Intervention: 38/50 • Usual care: 22/49 |
| | | | | | Positive EID result at 4-8wks: • Intervention: 0/38 • Usual care: 1/22 |
| Health systems improvement interventions | | | | | |
| [a]Finocchario-Kessler 2014 [30] | Kenya | Before and After study | 425 | Internet-based program that triggers electronic alerts by text messages to mothers' mobile phones | EID uptake at 4-8wks: • Intervention: 202/275 • Usual care: 99/150 |
| | | | | | Positive EID result at 4-8wks: • Intervention: 11/202 • Usual care: 7/99 |
| | | | | | Turnaround time of EID to caregiver: • Intervention: 3.5 (2.1–4.8) wks. • Usual care: 3.4 (2.1–5.0) wks. |
| | | | | | Turnaround time of EID results availability to mother: • Intervention: 1.3 (1.0–1.6) wks. • Usual care: 4 (2.3–10) wks. |
| | | | | | Initiation of ART by HIV-positive infant: • Intervention: 11/11 • Usual care: 1/7 |

(*Continued*)

**Table 1.** (Continued)

| Author & year | Country of intervention | Study design | Number of participants (Infants ≤ 8weeks of age) | Intervention(s) | Results of pre-specified outcome measures in included studies |
|---|---|---|---|---|---|
| Finocchario-Kessler 2018 [29] | Kenya | Cluster randomized controlled trial | 558 | HIV Infant Tracking System (HITSystem) | EID uptake at 4-8wks:<br>• Intervention: 329/329<br>• Usual care: 229/298 |
| | | | | | Positive EID result at 4-8wks:<br>• Intervention: 15/329<br>• Usual care: 1/229 |
| | | | | | Turnaround time of EID to caregiver:<br>• Intervention: 2.9wks. (IQR 2.2–4.6)<br>• Usual care: 5.5wks. (IQR 3.7–9.4) |
| | | | | | Turnaround time of EID results availability to mother:<br>• Intervention: 2wks. (IQR 1–3.4)<br>• Usual care: 3.3wks. (IQR 1.9–4.7) |
| | | | | | Initiation of ART by HIV-positive infant:<br>• Intervention: 15/15<br>• Usual care: 1/1 |
| Gupta 2016 [43] | India | Before and After study | 2770 | EID Follow-up system: a web-based tool that generated automated SMS and emails for reminding the field level staff | EID uptake at 4-8wks:<br>• Intervention: 638/1139<br>• Usual care: 1117/1631 |
| Herlily 2015 [42] | Zambia | Before and After study | 1059 | Service improvement: Three (3) components: (1) training of 132 ANC providers, (2) establishment of laboratory courier system to expedite CD4 results, and (3) follow-up of mother-infant pairs by 82 community-based lay counselors | EID uptake at 4-8wks:<br>• Intervention: 309/553<br>• Usual care: 202/506 |
| **Service integration interventions** | | | | | |
| Aliyu 2016 [35] | Nigeria | Cluster RCT | 320 | Integrated package of PMTCT services (Point-of-care CD4 cell count or percentage testing; decentralized PMTCT tasks to trained midwives (task shifting), integrated mother and infant care services, male partner participation, and community involvement | EID uptake at 4-8wks:<br>• Intervention: 125/150<br>• Usual care: 15/170 |
| | | | | | Positive EID result at 4-8wks:<br>• Intervention: 3/125<br>• Usual care: 5/15 |
| Washington 2015 [34] | Kenya | Cluster RCT | 1162 | Integration of ANC, PMTCT, and HIV care | EID uptake at 4-8wks:<br>• Intervention: 143/568<br>• Usual care: 106/594 |
| | | | | | Positive EID result at 4-8wks:<br>• Intervention: 6/143<br>• Usual care: 7/106 |
| **Behavioral interventions** | | | | | |
| Igumbor 2019 [38] | Kenya | Before and After study | 497 | Psychosocial support | EID uptake at 4-8wks:<br>• Intervention: 1117/1161<br>• Usual care: 992/1143 |
| Liu 2019 [36] | Nigeria | RCT | 449 | Conditional cash transfer | EID uptake at 4-8wks:<br>• Intervention: 63/195<br>• Usual care: 34/254 |
| Sam-Agudu 2017 [37] | Nigeria | Prospective cohort | 2304 | Mother-to-Mother (Mentor-Mother) support | EID uptake at 4-8wks:<br>• Intervention: 238/260<br>• Usual care: 170/237 |
| | | | | | Positive EID result at 4-8wks:<br>• Intervention: 1/238<br>• Usual care: 2/170 |

(*Continued*)

**Table 1.** (Continued)

| Author & year | Country of intervention | Study design | Number of participants (Infants ≤ 8weeks of age) | Intervention(s) | Results of pre-specified outcome measures in included studies |
|---|---|---|---|---|---|
| Weiss 2014 [41] | South Africa | Cluster RCT | 67 | Partner (Male) involvement | EID uptake at 4-8wks:<br>• Intervention: 30/30<br>• Usual care: 37/37 |
| | | | | | Positive EID result at 4-8wks:<br>• Intervention: 1/30<br>• Usual care: 3/37 |

Footnote:

a: Data from only the urban health facility.

The risk of bias graph (Fig 2) and the risk of bias summary (Fig 3) present bias in included studies. The effect sizes and quality of evidence for each outcome, according to GRADE, are presented as S4–S6 Tables.

## eHealth interventions versus usual care

Four RCTs [31–33, 44] and two observational studies [39, 40] compared eHealth interventions to usual care. Four studies compared short message systems (SMS) intervention with usual care [31, 32, 39, 44] and one study compared both SMS and telephone messaging to usual care [40], while another study compared telephone counseling to usual care [33].

There was no evidence that eHealth interventions increased uptake of EID at 4–8 weeks of age from four randomized trials of moderate risk of bias (OR 1.35, 95% CI 0.93 to 1.96, moderate certainty evidence: Fig 4, favors usual care). See S4 Table for GRADE quality of evidence.

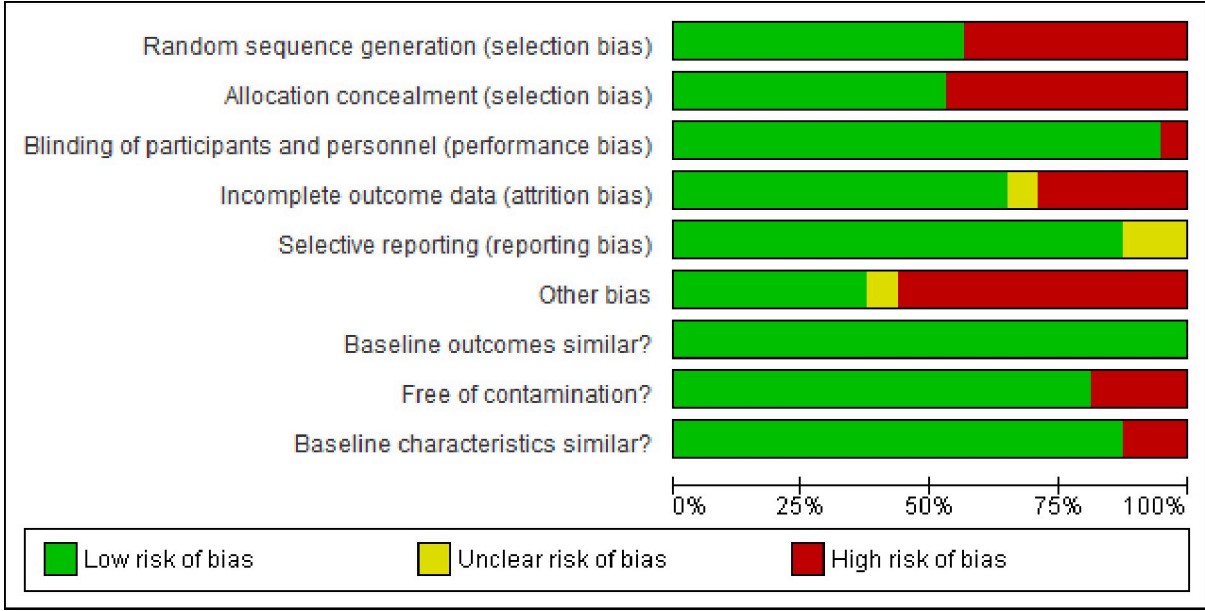

**Fig 2. Risk of bias graph.**

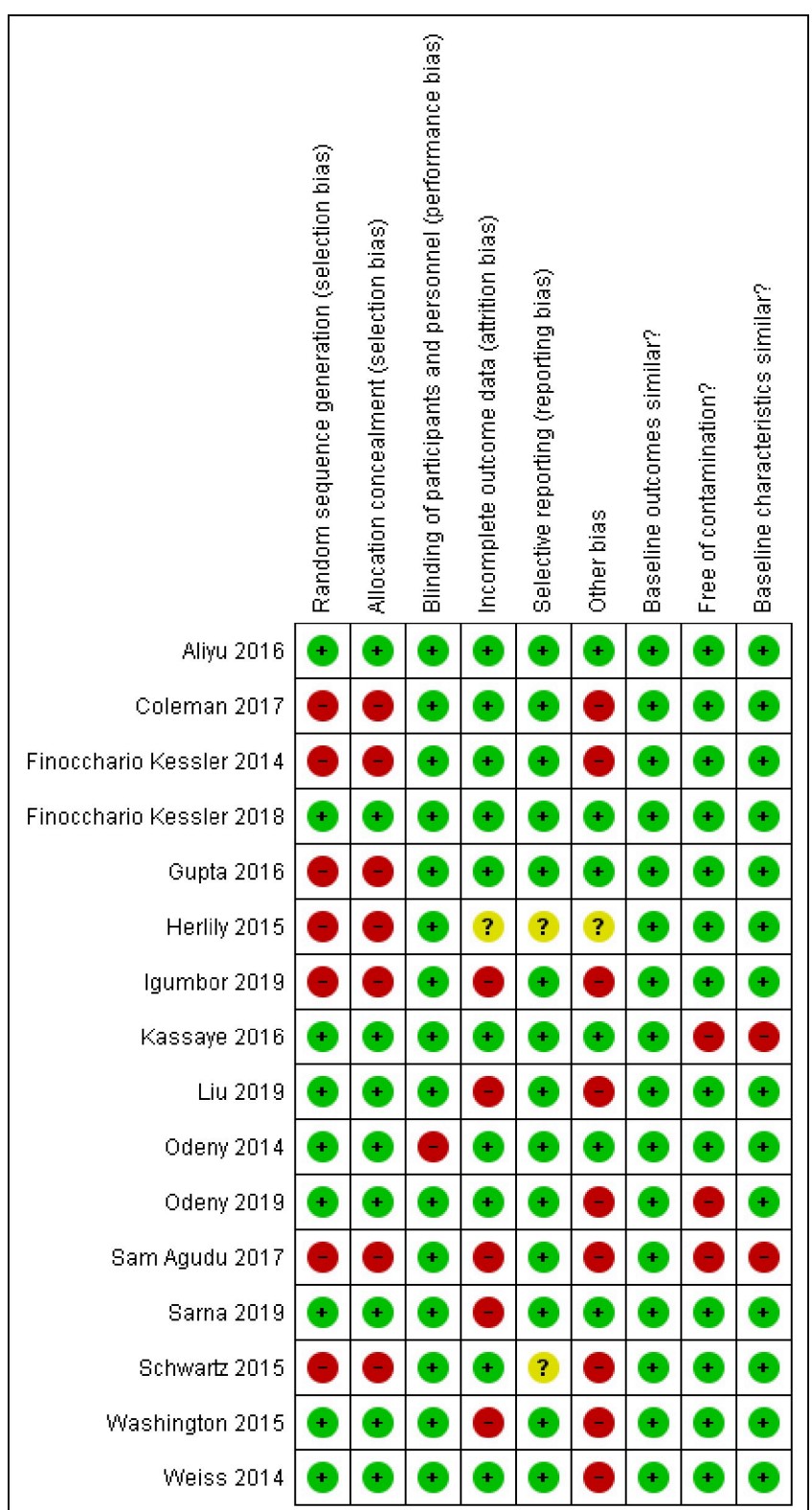

**Fig 3. Risk of bias summary: Review authors' judgements about each risk of bias for each included study.**

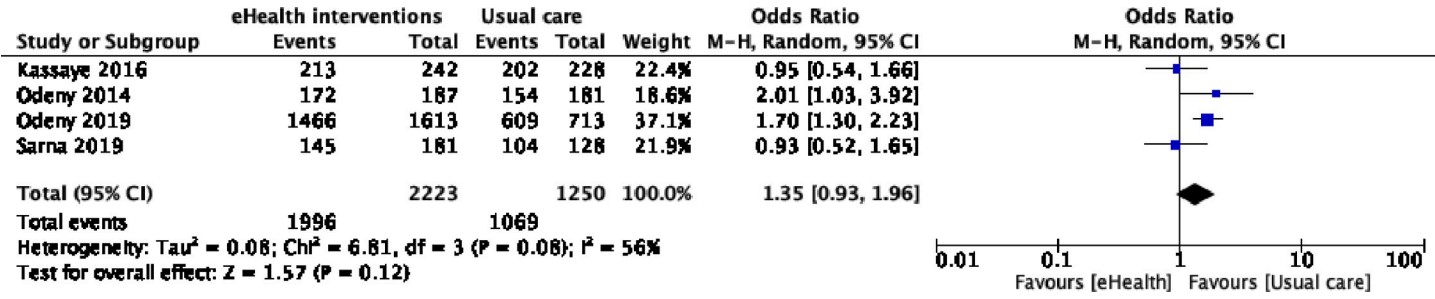

**Fig 4. eHealth interventions (RCTs) vs. usual care (uptake of EID at 4–8 weeks of age).**

In sensitivity analyses of individually randomized studies (OR 1.34, 0.63 to 2.85, Fig 5) and cluster-randomized studies (OR 1.34, 0.76 to 2.36, Fig 6), there was sustenance of lack of evidence that eHealth interventions were effective to increase uptake of EID at 4–8 weeks of age.

An individual RCT of "SMS only" intervention [44] had a point estimate of 2.10 (1.03.3.92) and did not indicate any evidence of SMS-only interventions increasing uptake of EID at 4–8 weeks age. A cohort study involving 639 HEIs has an effect estimate of 1.41 (0.93 to 2.16) [39], while a quasi-randomized study involving 99 HIEs has an effect size of 3.89 (1.65 to 9.18) [40]. Both non-experimental studies did not show that eHealth interventions increased uptake of EID at 4–8 weeks of age.

eHealth interventions did not increase the identification of HIV-infected infants; three randomized studies of moderate risk of bias [31, 33, 44] (OR 1.12, 95% CI 0.20 to 6. 35, moderate certainty evidence: Fig 7, does not favor intervention). Sensitivity analyses of only individually randomized trials (OR 1.84, 0.22 to 15.52, Fig 8) and SMS-only interventions (OR 0.45, 0.11 to 1.83, Fig 9) did not change the direction of effects.

A cohort study involving 493 HEIs had an effect size of 0.31 (0.02 to 5.95) [39], while a quasi-randomized study involving 60 HEIs has an effect size of 0.19 (0.01 to 4.77) [40]. Both non-randomized studies did not increase the identification of HIV-infected infants. The six studies [31–33, 39, 40, 44] for this comparison did not report data on any other review outcomes.

## Health systems improvement interventions versus usual care

One RCT [29] and three observational studies [30, 42, 43]. compared health systems improvement interventions to usual care. Two studies compared the HIV infant tracking (HIT) system (Internet-based automated alerts to providers and laboratory technicians to perform scheduled tasks and send results to mothers) [29, 30]. One study implemented a web-based SMS and email reminders to field-level program staff [43]. In contrast, the last study implemented a

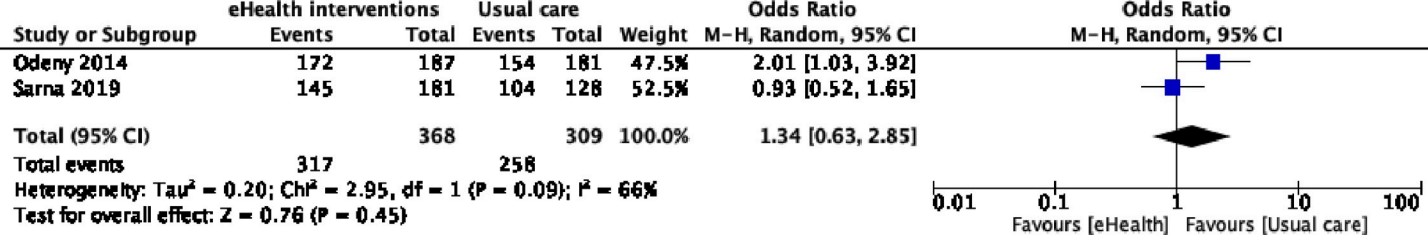

**Fig 5. Sensitivity analysis with individual RCT eHealth interventions vs. usual care.**

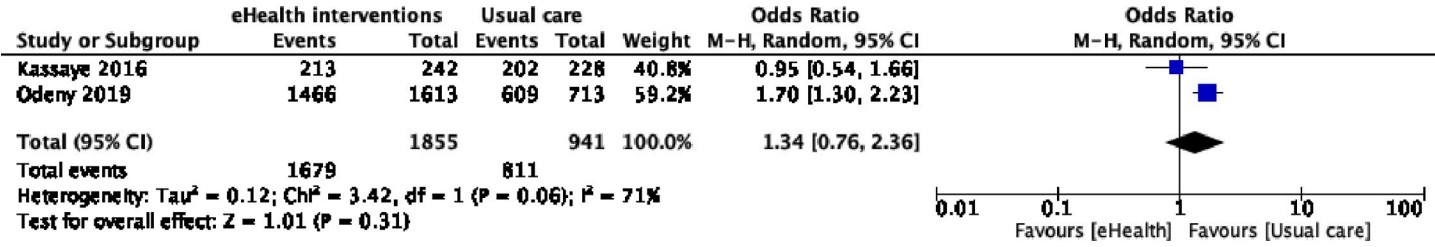

**Fig 6. Sensitivity analysis with cluster RCT eHealth interventions vs. usual care.**

combination of strategies (provider training, laboratory courier systems, and community follow-up of mother-infant pairs) [42].

A cluster-randomized study involving 690 HEIs has an effect size of 1.57 (1.08 to 2.30) [29].

Meta-analysis of data from three quasi-randomized studies with a high risk of bias [30, 42, 43] showed no evidence that interventions that improve health systems increase uptake of EID at 4–8 weeks of age (OR 1.12, 0.50 to 2.53, very low certainty evidence, Fig 10). The lack of evidence was sustained in a sensitivity analysis that excluded data from the largest study [43] (OR 1.67, 1.35 to 2.06, Fig 11, favors usual care). See S5 Table for GRADE quality of evidence.

Regarding identifying HIV-infected infants, a cluster RCT had an effect size of 10.89 (1.43 to 83.05) [29]. A quasi-experimental study had an effect size of 0.76 (0.28 to 2.02) and 99.67 (3.52 to 2818.12) for identification of HIV-infected infants and initiation of antiretroviral therapy, respectively [30]. Neither of the studies indicated evidence interventions to improve health systems increased the identification of HIV-infected infants.

The turnaround time of EID result to the caregiver was shorter at intervention facilities at 2.9 weeks [IQR 2.2–4.6]), compared to control facilities (median 5.5 weeks [IQR 3.7–9.4]) [29]. From an observational study [30], control facilities had similar turnaround time (median 3.4 weeks (IQR 2.1–5.0)) to intervention facilities (median 3.5 weeks (IQR 2.1–4.8)).

The turnaround time of EID results to mother was shorter (median of 2·0 weeks [IQR 1·0–3.4]) at intervention facilities, compared to control facilities (median 3.3 weeks [1.9–4.7]) [29]. An observational study reported a shorter turnaround time of results to mothers at intervention facilities (median 1.3 weeks (1.0–1.6), compared to control facilities (median 4.0 weeks (2.3–10.0)) [30]. No evidence indicates interventions to improve health systems increased the initiation of ART by HIV-infected infants. See S6 Table for the GRADE quality of evidence.

## Integration of EID versus usual care

Two included cluster RCTs with low risk [35] and high risk [34] of bias compared to integration of EID to other services. Each of these studies implemented a combination of strategies:

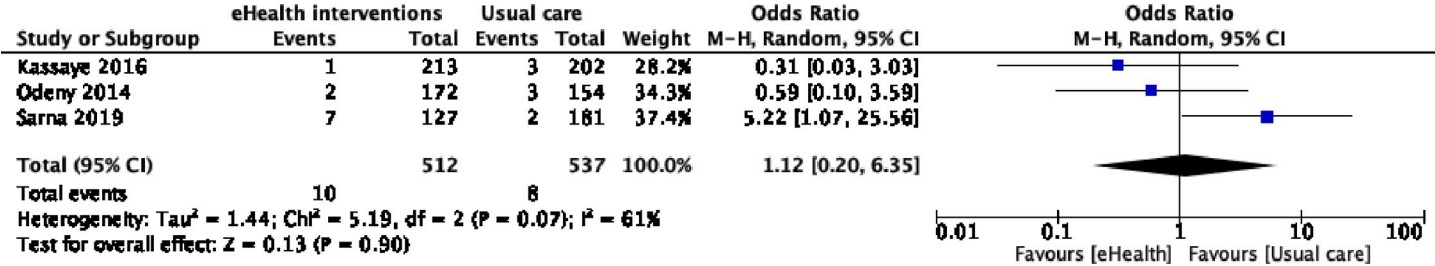

**Fig 7. eHealth interventions (RCTs) vs. usual care (identification of HIV-infected infants at 4–8 weeks of age).**

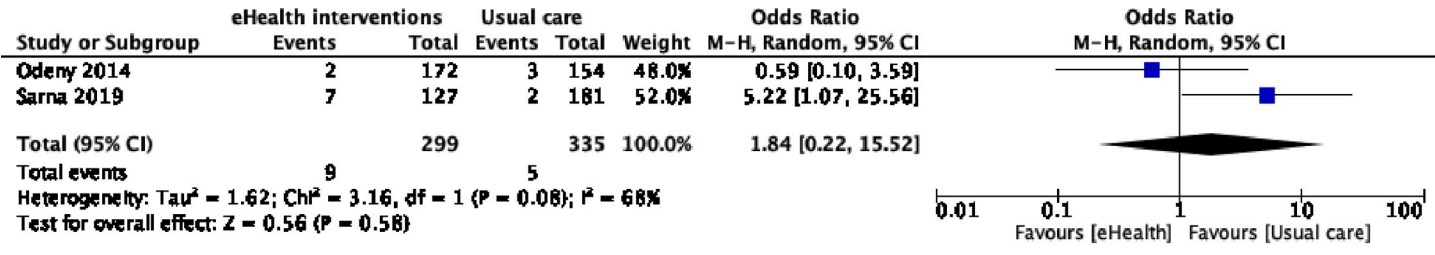

**Fig 8. Sensitivity analysis with individual RCTs for eHealth interventions (RCTs) vs. usual care (identification of HIV-infected infants at 4–8 weeks of age.**

integrated package of PMTCT services (Point-of-care CD4 cell count or percentage testing, decentralized PMTCT tasks to trained midwives (task shifting), integrated mother and infant care services, male partner participation, and community involvement) [35] and integration of antenatal, PMTCT, and HIV care [34].

Interventions that integrated EID services into other services, compared to usual care, did not increase uptake of early infant diagnosis of HIV at 4–8 weeks of age (OR 8.82, 95% CI 0.28 to 2818.12, very-low certainty evidence: Fig 12) [34, 35].

Very-low certainty evidence indicates interventions to integrate EID services into other services did not increase identification of HIV-infected infants (OR 0.19, 95% CI 0.02 to 2.28, very-low certainty evidence: Fig 13). See S6 Table for the GRADE quality of evidence. Neither study reported data on any other review outcomes [34, 35].

## Behavioral interventions

**Conditional cash transfer versus usual care.** In an RCT, conditional cash transfer compared with usual care had an effect estimate of 3.09 (1.93 to 4.94) and did not increase EID uptake at 4–8 weeks [36]. The study did not report data on any other review outcomes.

**Mentor-mother (mother-to-mother) versus usual care.** Two observational studies compared mentor-mother (mother-to-mother) intervention with usual care [37, 38]. Both studies implemented psychological support provided by HIV-positive mothers who had successfully gone through PMTCT strategies to mothers undergoing the strategies [37, 38].

A quasi-experimental study involving 497 infants had an effect estimate of 4.26 (2.53 to 7.17) [38], while a cohort study involving 2,304 HEIs had an effect estimate of 3.86 (2.73 to 5.46) [37]. There is no evidence from either study that mentor-mother support interventions, compared to usual care, increased EID uptake at 4–8 weeks of age. The studies did not report data on any other review outcomes.

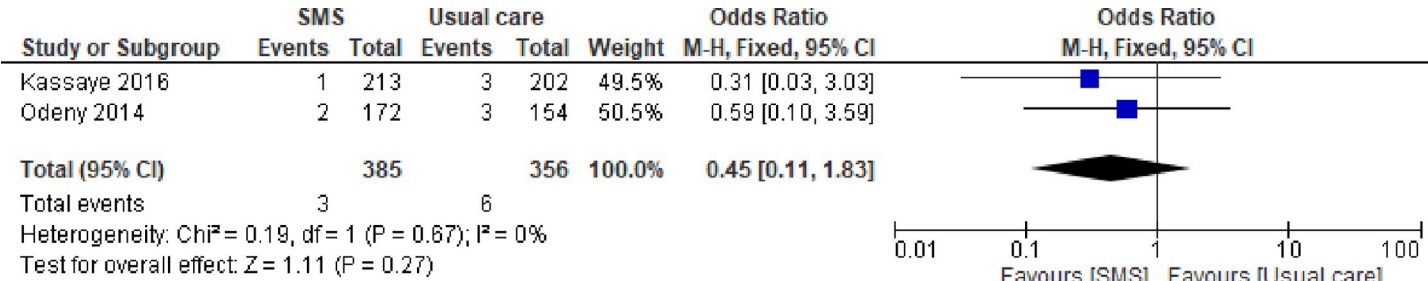

**Fig 9. Sensitivity analysis with "SMS only" interventions (RCTs) vs. usual care (identification of HIV-infected infant).**

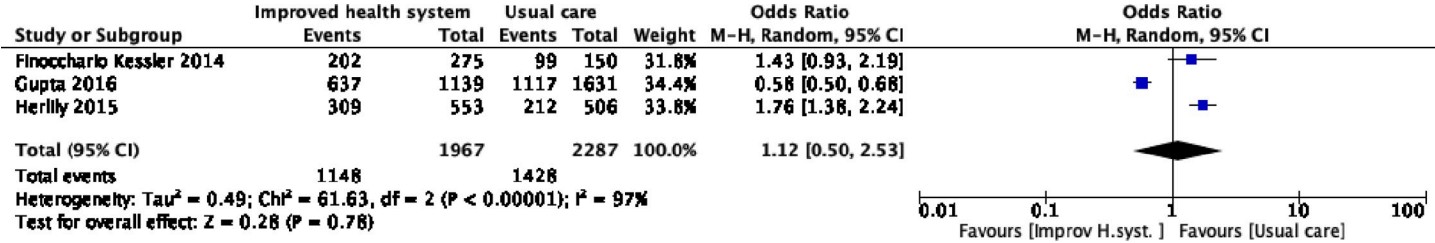

**Fig 10. Health systems improvement interventions vs usual care (uptake of EID at 4–8 weeks of age).**

**Male partner involvement versus usual care.** One RCT compared male partner involvement with usual care [41]. Compared to usual care, there is no evidence male partner involvement increased uptake of EID at 4–8 weeks of age (Effect size not estimable) or the identification of HIV-infected infants (OR 0.39 95% CI 0.04 to 3.96). The study did not report data on any other review outcomes.

We did not find an eligible study that implemented cognitive behavioral therapy interventions in comparison with usual care.

## Discussion

The objective of the systematic review was to identify strategies that effectively increased the uptake of EID of HIV infection at 4–8 weeks of age. The review included sixteen studies with low to high risk of bias conducted in SSA and India. Interventions evaluated include eHealth, health systems improvements, integration of EID into other services, conditional cash transfer, mother-to-mother support, and partner (male) involvement interventions. Compared to usual care, the review found no evidence that any of the evaluated interventions was effective to increase uptake of EID of HIV or increased identification of HIV-infected at 4–8 weeks of life. It is important to highlight some significant point estimates (odd ratio) with narrow confidence intervals that favored routine practices instead of the intervention. eHealth (1.35, 0.93 to 1.96) and health systems improvement (1.67, 1.35 to 2.06) interventions fall in this category. Other analyses showed that most interventions were not effective because of very wide confidence intervals that included 1; the line of no effect. The small event rates of uptake of EID and positive EID results at 4–8 weeks of life were responsible for the findings.

The need to identify strategies that are effective to increase uptake of EID of HIV has made researchers conduct systematic reviews on the topic. Two systematic reviews on interventions to promote PMTCT service delivery and retention of mother and infants in care were identified [11, 12]. In the review by Ambia and Mandala, phone calls or SMS reminders "showed a statistically significant increase in uptake of early infant diagnosis of HIV at 6–10 weeks of age" [11]. This review did not find such effectiveness with an analysis of data on SMS

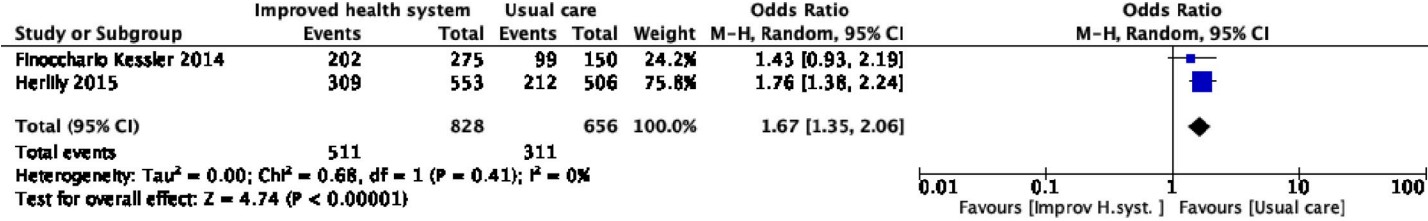

**Fig 11. Sensitivity analysis of health systems improvement interventions vs usual care with Gupta 2016 excluded.**

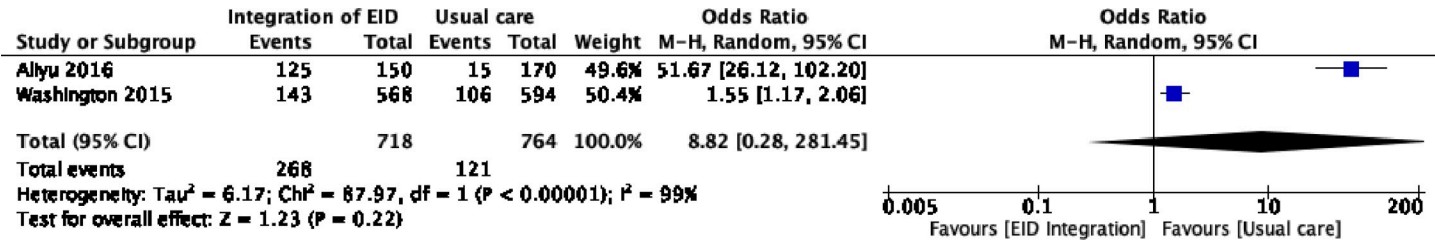

**Fig 12. Interventions that integrated EID services vs. usual.**

reminders. This might be because the review of Ambia and Mandala combined data of 2 RCTs and three observational studies in the analysis and included data on EID uptake up to 18 months of age [11]. For this review, meta-analyses were conducted based on study design, and data from observational and randomized trials were not meta-analyzed. Also, EID data up to 8 weeks of age were included because of the need for early HIV diagnosis in HIV-exposed infants. The other systematic review concluded "ANC/ART integration, family-centered approaches and use of lay health providers are demonstrably effective in increasing service uptake, and retention of HIV-positive mothers and their infants in PMTCT programmes" [12]. However, the review included a "Before and after" study that did not provide any pre-intervention data on EID despite the authors conclusions [12]. Hence, both systematic reviews had serious methodical flaws. From the foregoing, the findings of this systematic review did not agree with previous reports on effective interventions to increase EID uptake at 4–8 weeks or in the early diagnosis of new pediatric HIV infection.

Prevention of mother-to-child transmission of HIV infection is the global strategy for the elimination of new pediatric HIV infection, which is dependent on maternal viral load reduction to an undetectable level. EID diagnosis of HIV among exposed infants is recommended for 4–6 weeks of age or at the earliest opportunity after that [5]. HIV infection progresses very rapidly in infants if undetected, with 30% of undiagnosed infants dying before their first birth anniversary and 80% dying before the age of 5 years [19, 45]. EID allows for prompt antiretroviral therapy initiation and improves the survival of infants living with HIV. Although global EID uptake has increased to 59%, regional uptake lags—ranging from 29.3% in West and Central Africa to 68.8% in Eastern and Southern Africa [45]. This possibly explains the accommodation of HIE for testing, well beyond the recommended age limit of 8 weeks of age. Also, to improve the detection of HIV infection, birth testing has been implemented in specific settings as this model eliminates the need for the HEI to be returned for testing. However, this is not EID in the real sense.

Included studies were conducted in settings with a high HIV burden and significant contributions to new pediatric HIV infection [45]. This validates the search strategies for the review

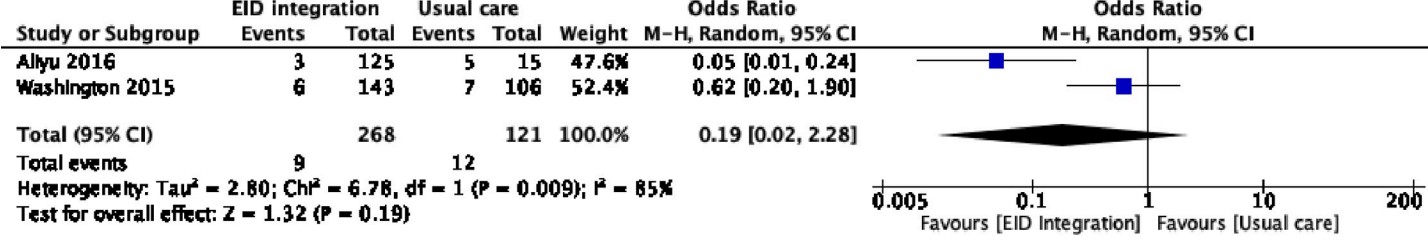

**Fig 13. Interventions that integrated EID services vs. usual care (identification of HIV-infected infant).**

and reduces the chances of omitting any eligible studies. The review used standard methods and the EPOC tool recommended for risk of bias assessment of public health interventions [26]. It provided evidence on each outcome using the GRADE approach [46]. The use of comprehensive search approach, inclusion of randomized and observational studies, and the removal restriction on language, geographic, or year of publication, ensured that this review is inclusive of currently available evidence on the subject, thus reducing helped minimized bias in the review process. Review authors, in pairs, screened titles for inclusion eligibility, data extraction, and risk of bias assessment. The use of random-effect meta-analysis in the review is justified by the inclusion of studies that evaluated multiple components of the same intervention, irrespective of being either randomized or observational studies. Finally, a strength of this review is the conduct of meta-analysis based on type of intervention implemented to increase EID uptake.

Despite the review's strength, it is limited by the dependence of EID uptake on women's entry into the PMTCT cascade of care during pregnancy. This might be responsible for the small sample sizes of many included studies. More so, included studies did not report the women's health status and whether the women had HIV virologic suppression, which has implications on vertical transmission of HIV infection. Hence, HIV testing in pregnancy should be scaled in a setting with high HIV prevalence and detailed health information recorded for each woman. Also, most of the included studies had small sample sizes and very few events, which led to wide confidence intervals of the effect estimates. The inclusion of non-randomized studies, though with good intention, led to studies with a high risk of bias and analyses with low-certainty evidence.

This review found no evidence of the effectiveness of some interventions previously said to be effective. With the investments in eliminating vertical transmission of HIV infection, careful consideration of future programmatic strategies to increase EID uptake is required. More experimental studies with many HEIs are needed to determine effective interventions to increase EID uptake. With increasing mobile-phone penetration in SSA and COVID-19 pandemic, eHealth interventions, including mobile phone interventions, should be further evaluated. Since many of the included studies had multiple components in their interventions, future studies should consider assessing the effectiveness of single interventions so that it is easy to determine which specific intervention is effective. Such intervention should be started early in pregnancy considering PMTCT interventions are implemented as a continuum of care once started for pregnant women living with HIV. As male partners often play essential decision-making roles, there is a need for male-partner involvement to be evaluated in large studies. While conditional cash transfer interventions are likely to increase EID uptake because of the financial benefits to the mother, it is not a sustainable intervention. It might be counterproductive in the long run. The effect of point-of-care (POC) testing and birth testing on EID uptake is uncertain as HEIs would have presented for testing before being offered POC test, and birth testing might not require a return for testing. The two testing mechanisms do not deal with the challenge of returning an HEI for EID. However, interventions to evaluate birth testing might include repeat testing at 4–8 weeks for infants with negative HIV tests at birth. Until an effective strategy is identified, usual care at all settings should be strengthened to increase EID.

## Conclusion

This systematic review compared common interventions with usual care and found no evidence that any intervention was effective in increasing EID uptake at 4–8 weeks of age or increasing the identification of HIV-infected infants. Also, there is no evidence that any of the

evaluated interventions increased ART initiation and turnaround time of results to the care-giver and mother. More large studies are required to identify effective strategies that increase EID uptake.

## Supporting information

**S1 Table. Search strategies.**
(DOCX)

**S2 Table. Table of excluded studies.**
(DOCX)

**S3 Table. Characteristics of included studies.**
(DOCX)

**S4 Table. Quality of evidence for ehealth interventions vs. usual care for randomized controlled trials.**
(DOCX)

**S5 Table. Quality of evidence for interventions to improve health systems vs. usual care for observational studies.**
(DOCX)

**S6 Table. Quality of evidence for interventions to integrate EID services vs. usual care.**
(DOCX)

## Acknowledgments

We acknowledge the contribution of Jean McClelland, Librarian at the Mel and Enid Zuckerman College of Public Health, the University of Arizona, in finalizing the search strategies and in conducting the database search.

## Author Contributions

**Conceptualization:** Babasola Okusanya, John Ehiri.

**Data curation:** Linda J. Kimaru, Namoonga Mantina.

**Formal analysis:** Babasola Okusanya, Linda J. Kimaru, Namoonga Mantina, Lynn B. Gerald, Sydney Pettygrove.

**Project administration:** Babasola Okusanya, John Ehiri.

**Supervision:** Lynn B. Gerald, Sydney Pettygrove, Douglas Taren, John Ehiri.

**Writing – original draft:** Babasola Okusanya.

**Writing – review & editing:** Linda J. Kimaru, Namoonga Mantina, Lynn B. Gerald, Sydney Pettygrove, Douglas Taren, John Ehiri.

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
