## [Decision Letter · Decision Letter 0]

23 Aug 2021

PONE-D-21-16973

Interventions to increase early infant diagnosis of HIV infection: a systematic review and meta-analysis

PLOS ONE

Dear Dr. Okusanya,

Thank you for submitting your manuscript to PLOS ONE. After careful consideration, we feel that it has merit but does not fully meet PLOS ONE’s publication criteria as it currently stands. Therefore, we invite you to submit a revised version of the manuscript that addresses the points raised during the review process.

ACADEMIC EDITOR: This is an interesting analysis of interventions to increase uptake of early-infant diagnosis. The authors review the existing data and present a meta analysis. The manuscript addresses an ongoing problem but it lacks clarity in some of the methodology, results reporting, and conclusion. The authors will need to address the comments by reviewer #2 before the manuscript could be accepted for publication. In particular the authors should clarify the personnel that conducted the review - it lists both 2 and 3 different individuals. The authors should also justify their meta analysis by study design. For the results, I agree with reviewer 2 that it is not clear what represents summary of the primary authors results vs meta-analysis. This should be clarified. Table 1 should be modified to include the reference, number of participants and primary results. IT is unclear how the authors reach their conclusion that no intervention increases EID when several study designs appear to significantly increase EID. The authors should discus how bias in study design can alter theses results. 

The authors should also edit the manuscript for grammar, punctuation, and syntax throughout. The authors should also use person first language throughout (i.e. living with HIV instead of HIV positive or HIV infected).

We look forward to receiving your revised manuscript.

Kind regards,

Brian C. Zanoni, MD

Academic Editor

PLOS ONE

1. Please ensure that your manuscript meets PLOS ONE's style requirements, including those for file naming. The PLOS ONE style templates can be found at https://journals.plos.org/plosone/s/file?id=wjVg/PLOSOne_formatting_sample_main_body.pdf and https://journals.plos.org/plosone/s/file?id=ba62/PLOSOne_formatting_sample_title_authors_affiliations.pdf.

2. Please note that PLOS ONE uses a single-blind peer review procedure. We would therefore be grateful if you could include in the information that has been redacted for peer review in the manuscript.

Additional Editor Comments (if provided):

This is an interesting analysis of interventions to increase uptake of early-infant diagnosis. The authors review the existing data and present a meta analysis. The manuscript addresses an ongoing problem but it lacks clarity in some of the methodology, results reporting, and conclusion. The authors will need to address the comments by reviewer #2 before the manuscript could be accepted for publication. In particular the authors should clarify the personnel that conducted the review - it lists both 2 and 3 different individuals. The authors should also justify their meta analysis by study design. For the results, I agree with reviewer 2 that it is not clear what represents summary of the primary authors results vs meta-analysis. This should be clarified. Table 1 should be modified to include the reference, number of participants and primary results. IT is unclear how the authors reach their conclusion that no intervention increases EID when several study designs appear to significantly increase EID. The authors should discus how bias in study design can alter theses results.

Reviewers' comments:

Reviewer's Responses to Questions

**Comments to the Author**

1. Is the manuscript technically sound, and do the data support the conclusions?

Reviewer #1: Yes

Reviewer #2: Partly

2. Has the statistical analysis been performed appropriately and rigorously? 

Reviewer #1: Yes

Reviewer #2: I Don't Know

3. Have the authors made all data underlying the findings in their manuscript fully available?

Reviewer #1: Yes

Reviewer #2: No

4. Is the manuscript presented in an intelligible fashion and written in standard English?

Reviewer #1: Yes

Reviewer #2: Yes

5. Review Comments to the Author

Reviewer #1: The authors manuscript covers a very important topic in the paediatric HIV arena, especially in the low resource settings countries such as Sub Saharan Africa and India. This therefore means the interventions studied, which are within reach of low income countries, are compared systematically that can be very helpful in programme implementation and optimisation. As a practising clinician the findings of this review go a long way in helping many optimise their early infant diagnosis programmes. They write in vary clear, easy to follow fashion, in unambiguous language, detailed methodology that allows for others to potentially repeat their study. They have also review considerable amount of literature and are very thorough. No ethical or methodical concerns arise. Their conclusions are logical and based on their findings so I recommend their manuscript to be accepted.

Reviewer #2: This is an important study given the global prevalence of children living with HIV of which a significant proportion are not aware of their diagnosis and not currently on treatment. The authors attempt to address the critical opportunities to identify infants living with HIV early that are being missed by critically reviewing and analyzing current published interventions aimed to increase the uptake of early infant diagnosis (EID). The study provided a review of the current literature, highlighting 16 studies published between 2014 and 2019 that evaluated interventions involving eHealth services, service improvement projects, service integration programs, behavioral interventions, and male partner involvement. Additionally, authors conducted a meta-analysis of each of these different intervention types to identify interventions that increase EID of HIV at 4-8 weeks of age.

Overall, the systematic review and meta-analysis is well done. The major strengths of this manuscript are the focus on a population (HIV exposed infants) that has high rates of morbidity and mortality if undiagnosed and not linked to care early in life, as well as the extensive search and selection of studies. However, a few additional points to consider, including unclear meta-analysis methodology, additional information on initial studies (including in table 1), and further clarification of conclusions based on evidence presented. I would also consider minor revisions including the use of identity-first vs person-first language, revising for punctuation and grammar throughout, incorrect formatting of methods section, and lack of appendices 4-6 included in submission supplementary materials.

1. Within the methods section, it is unclear how many authors reviewed results and extracted data. The authors state two authors reviewed them, but list three different people (lines 141, 146, and 153). Please clarify how many authors reviewed/extracted data.

2. Additional concern with the methodology is regarding the meta-analysis. Specifically explain why you chose to conduct the meta-analysis by intervention type and define measures utilized, including the effect estimate. It is also not clear which studies you included in the meta-analysis.

3. Between the methods and results section, the blank spaces between lines 190-201 need to be removed/reformatted.

4. A concern within the results section is that it is unclear what is a summary of the individual study results versus what are the results of the author’s meta-analysis. The authors could consider presenting a summary of the results as reported by the primary study authors and then the results of their meta-analysis by each intervention sub-type. Authors could also provide a table with the cumulative data that each meta-analysis is based on to better support their meta-analysis results.

5. An additional concern with the results is summary of information included in Tables 1 and 3. Authors should consider re-organizing Tables 1 and 3 to mirror results section (e.g., organized by intervention type with subheadings as opposed to study design). Additionally, Table 1 should provide additional useful information about each study, including number of participants and study results. Table 1 should also include references so that readers can easily refer to cited studies. Also it is unclear what the serial number in the tables are for and authors could consider removing these.

6. Within the appendices, figure 1 arrowheads at the top of the PRISMA flow diagram are incorrect. Authors should fix the formatting of the arrows.

7. An additional concern is the authors conclusion within the discussion. It is unclear how authors came to the conclusion that there was no evidence that any of the evaluated interventions increased uptake of EID compared to usual care based on the evidence you present in the results section, including statistically significant effect estimates/OR’s showing what appears to be uptake in EID.

8. I am concerned with the authors use of identity-first language in throughout the manuscript. The authors should use person-first language throughout the manuscript. For example: “HIV infected infants” can be changed to “infants living with HIV.”

9. Authors should revise for punctuation and grammar throughout the manuscript.

10. Authors refer to appendices 4-6, however these were not provided in the supplementary materials.

6. PLOS authors have the option to publish the peer review history of their article (what does this mean?). If published, this will include your full peer review and any attached files.

Reviewer #1: **Yes: **Mogomotsi Matshaba

Reviewer #2: No

---

## [Author Response · Author response to Decision Letter 0]

9 Sep 2021

Responses to reviewer #2 comments and that of the Academic Editor have been attached.

---

## [Decision Letter · Decision Letter 1]

7 Oct 2021

Interventions to increase early infant diagnosis of HIV infection: a systematic review and meta-analysis

PONE-D-21-16973R1

Dear Dr. Okusanya,

We’re pleased to inform you that your manuscript has been judged scientifically suitable for publication and will be formally accepted for publication once it meets all outstanding technical requirements.

Kind regards,

Brian C. Zanoni, MD

Academic Editor

PLOS ONE

Additional Editor Comments (optional):

The authors have sufficiently responded to the reviewer's comments.

Reviewers' comments:

Reviewer's Responses to Questions

**Comments to the Author**

1. If the authors have adequately addressed your comments raised in a previous round of review and you feel that this manuscript is now acceptable for publication, you may indicate that here to bypass the “Comments to the Author” section, enter your conflict of interest statement in the “Confidential to Editor” section, and submit your "Accept" recommendation.

Reviewer #1: All comments have been addressed

Reviewer #2: All comments have been addressed

2. Is the manuscript technically sound, and do the data support the conclusions?

Reviewer #1: Yes

Reviewer #2: Yes

3. Has the statistical analysis been performed appropriately and rigorously? 

Reviewer #1: Yes

Reviewer #2: Yes

4. Have the authors made all data underlying the findings in their manuscript fully available?

Reviewer #1: Yes

Reviewer #2: Yes

5. Is the manuscript presented in an intelligible fashion and written in standard English?

Reviewer #1: Yes

Reviewer #2: Yes

6. Review Comments to the Author

Reviewer #1: Accept with a few minor edits:

line 143 : Close the inverted commas at 143. Rest of the paper is clear and acceptable for publication.

Reviewer #2: The authors have provided a nicely detailed and thorough response to the comments from the previous review and have addressed my concerns. The study findings are important and will be of interest to a broad audience. Would ensure that font and formatting is all uniform, as there is different size font in different sections of the manuscript. Additionally, a comment about parenthesis (table 1) is included in the "clean" version of the revised manuscript and should be removed.

7. PLOS authors have the option to publish the peer review history of their article (what does this mean?). If published, this will include your full peer review and any attached files.

Reviewer #1: **Yes: **Mogomotsi Matshaba

Reviewer #2: No

---

## [Editor Report · Acceptance letter]

8 Feb 2022

PONE-D-21-16973R1 

Interventions to increase early infant diagnosis of HIV infection: a systematic review and meta-analysis 

Dear Dr. Okusanya:

I'm pleased to inform you that your manuscript has been deemed suitable for publication in PLOS ONE. Congratulations! Your manuscript is now with our production department. 

Kind regards, 

on behalf of

Dr. Brian C. Zanoni 

Academic Editor

PLOS ONE